# Genetic Stability of Mesenchymal Stromal Cells for Regenerative Medicine Applications: A Fundamental Biosafety Aspect

**DOI:** 10.3390/ijms20102406

**Published:** 2019-05-15

**Authors:** Simona Neri

**Affiliations:** Laboratorio di Immunoreumatologia e Rigenerazione Tissutale, IRCCS Istituto Ortopedico Rizzoli, 40136 Bologna, Italy; simona.neri@ior.it; Tel.: +39-0516-3668-03

**Keywords:** mesenchymal stem/stromal cells, genetic stability, senescence, tumorigenicity, biosafety

## Abstract

Mesenchymal stem/stromal cells (MSC) show widespread application for a variety of clinical conditions; therefore, their use necessitates continuous monitoring of their safety. The risk assessment of mesenchymal stem cell-based therapies cannot be separated from an accurate and deep knowledge of their biological properties and in vitro and in vivo behavior. One of the most relevant safety issues is represented by the genetic stability of MSCs, that can be altered during in vitro manipulation, frequently required before clinical application. MSC genetic stability has the potential to influence the transformation and the therapeutic effect of these cells. At present, karyotype evaluation represents the definitely prevailing assessment of MSC stability, but DNA alterations of smaller size should not be underestimated. This review will focus on current scientific knowledge about the genetic stability of mesenchymal stem cells. The techniques used and possible improvements together with regulatory aspects will also be discussed.

## 1. Introduction

Human stem cells include three main cell types: pluripotent embryonic stem cells (ESCs) [1] human induced pluripotent stem cells (iPSCs) generated from adult somatic cells reprogrammed to a pluripotent state by introduction of specific transcription factors [2]; multipotent adult stem cells, with reduced differentiation and proliferation ability compared to the previous two [3]. The latter group includes mesenchymal stem/stromal cells (MSCs), a heterogeneous group of tissue-specific cells, with no unique cluster of differentiation (CD) signature, that can be isolated from many tissues [4].

MSCs were identified for the first time as fibroblastic colony-forming units (CFU-Fs) in the bone marrow (BM) [5]. At present, there is no consensus on a single surface molecule to identify MSCs and the International Society for Cellular Therapy (ISCT) defined minimal criteria for their characterization [6].

MSCs appear to be extremely powerful tools for tissue engineering and regenerative medicine, not just for their multilineage differentiation potential (MSC engraftment after in vivo delivery is low [7]) but mostly for their paracrine activity, exerted through the release of soluble factors [8] such as anti-inflammatory cytokines, anti-apoptotic and trophic molecules that stimulate tissue repair and counteract inflammation. These factors are able to modify injured tissue microenvironment [9,10]. More recently, a substantial contribution to tissue regeneration through exosome and microvesicle release has been described [11]. MSCs show homing ability to sites of inflammation, injury and tumor. Their immunomodulatory properties [9,12] can be useful to treat immune-mediated diseases and graft-versus-host disease [13,14]. In addition, these cells can be harvested using minimally invasive procedures from many different adult tissues and can be easily manipulated in vitro, as for in vitro expansion and differentiation. Finally, their derivation from adult tissues solves ethical challenges represented by the use of other cell types such as embryonic stem cells. For all these reasons these cells are an ideal candidate for cellular therapies and tissue engineering offering groundbreaking new opportunities for the treatment of disease and injury, with an enormous potential for development.

MSCs are available from many human adult tissues [15]. In accordance with the hypothesis of their perivascular origin, they are present virtually in all vascularized tissues. Actually, human perivascular cells sorted from diverse human tissues and cultured over the long term give rise to adherent, multilineage progenitor cells that exhibit the features of MSCs [16]. The most commonly used MSC sources are bone marrow (BM) [17] and adipose tissue (AT) [18,19,20,21]. They can also be isolated from a series of other tissues like umbilical cord (UC) or Wharton’s Jelly [22], cord blood [23,24], placenta and amnion [25,26], amniotic fluid [27], chorionic villi [28], dental pulp [29,30], endometrium [31], brain [32], skin [33], synovial membrane and synovial fluid [15,34], sweat glands [35], nasal mucosa [36].

Despite several common aspects, the biological characteristics and differentiation abilities of MSCs are a function of the tissue of origin [37]. Moreover, by whole transcriptome sequencing analysis (RNA-seq), it was shown that MSC gene expression and phenotype are also strongly dependent on the substrate [38], in particular on substrate chemistry and topography, thus highlighting the relevance of the interactions between cells and biomaterials [39,40].

Mesenchymal stem cell research exploded in the last years and has now reached all areas of biology, including tissue regeneration, modeling of complex tissues by juxtaposing independently patterned cellular components, developmental epigenetic regulation, and cancer [41,42]. Progresses in the clinical use of MSCs came from the combination of stem cells and tissue engineering techniques with the use of scaffolds and biomaterials to improve the therapeutic efficacy of MSCs alone [43].

MSCs are at present largely used in clinical trials: to end of March 2019, 6071 studies on “Adult stem cells”, 945 on “mesenchymal stem cells” and about 190 on “mesenchymal stromal cells” were registered in the public clinical trials database of the US National Institutes of Health, of which 2356, 263 and 47, respectively, completed. Most of them are phase I (safety) and II (efficacy) or combined Phase I/II interventional studies; both with allogeneic and autologous cells. The prevalent conditions for MSCs applications are represented by musculoskeletal, autoimmune, vascular, and central nervous system diseases, wounds and injuries (available online: https://clinicaltrials.gov). In addition, a number of nonregistered studies and trials are performed in many countries.

The aim of this review is to provide the reader with the current knowledge about genetic stability of mesenchymal stem cells for clinical use in relation to in vitro expansion and cellular senescence. The techniques used, their limitations and possible future improvements, together with regulatory aspects will also be discussed by a scientific point of view.

## 2. MSC Clinical Use

Despite a relatively still incomplete scientific knowledge about stem cell differentiation, transplantation and tissue integration [44], MSCs are already an expanding reality in the clinic due to their enormous therapeutic potential. Several studies provided promising results but further validation is needed and issues related to MSC general nature and biological behavior cannot be underestimated [45]. Actually, some studies showed varying outcomes, likely because of the differences in cell isolation and culture conditions, that could result in the selection of MSC subpopulations [46]. MSCs are heterogeneous populations, with phenotypic and functional features strongly dependent on the donor, the harvest site, and the culture conditions. Current culture expansion methods do not guarantee the preservation of native MSC properties causing variable quality and potency and accounting for inconsistent outcomes. Inconsistencies and variability in MSC studies may also arise from interindividual variability, the tissue of origin, intrinsic differences in cell-based products, lack of standardization. Efforts are therefore currently performed to standardize culture and differentiation conditions. Research aims at increasing clinical reproducibility by thorough characterization of cell heterogeneity [29] and by method standardization and development (as cell functionalization or priming to enhance therapeutic applications) [47]. One of the critical aspects contributing to variable results is MSC heterogeneity; the presence of different subpopulations with nonidentical characteristics in MSC preparations limits stem cell production methodology [4,18,29,48,49]. Heterogeneity might partly explain the variable clinical performance of the cells and the inconsistent results available in the literature, even if some authors claim that clonal heterogeneity plays a positive role on genetic stability keeping culture away from transformation [29]. By multicolor lentiviral barcode labeling and deep sequencing, the heterogeneous composition of umbilical cord-derived MSC preparations was shown to undergo dynamic clonal selection in culture, with an initial massive reduction in diversity and a selection of single clones over time, starting in the early passages [49]. Subset selection and cell priming have been proposed to overcome these disadvantages [47]. Another way to reduce heterogeneity is the use of clonal MSCs as a starting material, in order to maximize the homogeneity of the final stem cell products. This was performed by Yi et al. [48] by the development of a GMP-compatible procedure for clinical-grade production of human bone marrow-derived clonal MSCs obtained by subfractionation culturing. The drawback is the need for a massive cell expansion to obtain the desired number of cells, even if the authors demonstrate no acquisition of genetic alterations [48]. For some authors, manufacturing systems will hardly reach a unique standard due to the intrinsic differences among samples and experimental conditions. They suggest to concentrate efforts on standardizing methods for MSC characterization and potency evaluation to establish release criteria for MSC manufacturing for each specific experimental setting [50].

## 3. The Safety Issue

The clinical use of MSCs raises the fundamental concern of their safety. Several aspects concerning MSCs safety are still incompletely explored, therefore it is essential to minimize risks and that solid evidence of preclinical efficacy and safety is guaranteed.

MSC frequency in human adult tissues is low: approximately 1/10^6^ cells in adult bone marrow and a 100–1000-fold higher frequency in the umbilical cord and adipose tissue [51]. Due to the limited presence of MSCs in human tissues, ex vivo expansion is frequently required to obtain the sufficient number of cells to be administered, but this procedure has some drawbacks, thus raising the problems of genetic stability, senescence and transformation.

In vitro expansion reduces replicative potential and multipotency; it drives senescence; it reduces DNA polymerase and DNA repair efficiencies, thus leading to DNA damage accumulation, such as cytogenetic alterations (deletions, duplications, rearrangements), mutations, epigenetic changes [52,53,54,55,56,57,58,59]. This applies also to stem cells [60] (for a review on stem cell DNA damage and aging see Behrens et al. [61]). There are two possible outcomes of DNA damage: erroneous repair and persistent DNA damage; the first can lead to transformation, the second can block transcription and replication driving the aging process [62]. DNA damage is strictly related to aging and cancer and regulation of DNA damage checkpoints has a critical role in accelerating or decelerating tissue-aging and age-related carcinogenesis [56]. If on one side DNA damage accumulation can lead to senescence, on the other side it can induce genomic instability with a consequent increased risk of transformation. Moreover, cellular senescence can prevent/promote tumorigenesis [63]. MSC preparations, therefore, need a careful check of genetic alterations possibly developed or expanded in culture before their use in the clinic.

In general, MSC preparations ultimately enter senescence and stop proliferation, but transformation cannot be excluded. Actually, ex vivo expansion implies very high proliferation rates in an artificial environment lacking the mechanisms that guarantee the negative selection/clearance of altered cells such as those active in the whole organism. Genomic instability could arise from replicative stress imposed by in vitro culture and from chromosome condensation defects as those described for human pluripotent stem cells, showing higher instability compared to MSCs [64]. Therefore, multiple replications in vitro expose cells to the risk of accumulating genetic and epigenetic alterations, with a detrimental effect on cell biology and therapeutic properties, with the promotion of cell senescence and potential transformation, thus possibly affecting treatment efficacy and patient safety (Figure 1). Indeed, MSC exposure to DNA damaging agents reduces significantly the differentiation potential [65].

Several studies were performed to assess if culturing does alter cell phenotype and gene expression in relation to cellular ability to differentiate. A series of papers was published that demonstrate a decrease in proliferative, differentiating, homing, immunomodulatory properties and the development of a senescence-associated secretory phenotype, SASP, [57,58,66,67] with in vitro MSC aging. Conversely, a recent study using adipose-derived stem cells (ASCs) [68] demonstrated by RNA sequencing that cells maintain differentiation potential and consistent profile of key mesenchymal markers in long-term cultures, but associated with distinct RNA isoforms.

In clinical trials, cell culture usually lasts to the 2nd passage but, in some cases, more passages are needed. When large scale production is required and extensive expansion is performed, manufacturing processes integrating scientific knowledge and clinical perspectives with prospected industrial development are needed [50].

## 4. Senescence

MSC aging is accompanied by a series of genetic, epigenetic, transcriptional and translational changes affecting cell function [60,69] and is a critical aspect to be considered for cellular therapy and safety assessments [70]. Both in vivo [61] and in vitro aging should be considered. During in vitro expansion, proliferation rate progressively decreases until it reaches a senescent state with growth arrest [57,58]. MSCs show low or negative levels of telomerase activity [33,55,71] and their telomeres progressively shorten with in vitro age [57,72,73,74] but without affecting cultures until several passages are performed [55,75]. Together with telomere shortening and proliferation slowdown, differentiation ability is reduced [57,76], particularly in the case of adult-derived cells compared to child-derived [76]. The effect of donor age is supported by the observation that human umbilical cord MSCs, having a neonatal origin, can be grown for up to passage 10 without losing multipotential mesenchymal progenitors [77] and show increased telomerase activity and longer telomeres compared to adult BM-MSCs [78]. Senescence also affects the composition of MSC secretome [67]. Cell cycle arrest usually requires about 20–30 cell divisions, depending on donor age, tissue source and culture conditions [57,79]. On the contrary, conventional surface markers do not change [80,81]. The protein p16 is upregulated during in vitro aging with an accompanied demethylation of its promoter [58]. Furthermore, hypoxic conditions seem to delay/bypass senescence through downregulation of the cyclin-dependent kinase inhibitors p16 [82] and p21 [83] and upregulation of DNA repair genes with a concurrent increased genomic stability [84].

Epigenetics has recently gained a great deal of attention [85]. Epigenetics is not only involved in gene regulation and chromatin structure, but also in senescence and genomic stability [86,87,88]. Major concerns still to be addressed are the comprehensive evaluation of dynamic changes of the epigenetic marks and their effect on senescence, commitment and stemness [88]. Epigenetic anomalies have been hypothesized to potentially cause/influence MSC genetic stability and differentiation capacity. Phermthai et al. [27] investigated global genomic methylation and genetic imprinting of some imprinted genes during prolonged in vitro culture of amniotic fluid stem cells and found epigenetic instability in high passage (P) cultures (after P8) correlating with loss of differentiation potential. The expression levels of the analyzed imprinted genes gradually increased with culture time and parental allele-specific imprinting was found to be frequently lost in association with altered CpG methylation.

The principal epigenetic modifications of the genome are chromatin methylation at CpG sites and histone modifications (acetylation, methylation). Histone H3/H4 acetylation is significantly reduced with advancing culture passages in association with enhanced global histone deacetylase activity [87,89]. In general, there is a gradual decrease in global DNA methylation with MSC aging in culture [73,90]. Redaelli et al. [73] speculate that all the changes observed during culture (telomere shortening and loss of genomic regions) belong to a definite program regulated by epigenetic modifications. Hypomethylated regions (enriched in genes related to morphogenesis) prevail over hypermethylated ones (enriched in genes associated with differentiation) [86]. Moreover, methylation changes observed in long-term cultures of human MSCs overlapped those observed in vivo [91]. In vitro replicative senescence shares many aspects with physiological aging, with epigenetic modifications influencing gene expression profile and reducing stemness and the total number of MSCs [86]. Very interestingly, it was described that human aging is associated to very reproducible DNA methylation patterns with specific age-associated CpG sites whose detection can constitute a biomarker for chronological age estimation, a sort of “epigenetic clock” [92]. Similarly, in vitro replicative senescence is accompanied by DNA methylation changes tightly regulated and reproducible [93,94,95] that can be used as a biomarker to evaluate aging of cell preparations for quality controls. An “epigenetic-senescence-signature” based on six CpGs whose methylation level correlates with the number of population doublings was recently described in human BM-MSCs [96,97]. However, these CpG methylation profiles significantly differed among subpopulations of the same MSC preparation, indicating that the epigenetic state of cellular senescence within MSC preparations is highly heterogeneous and these modifications are not synchronous [98]. In addition to 5-methylcytosine (5mC), 5-hydroxymethylcytosine (5hmC) was recently described as an epigenetic mark possibly related to cellular aging. Hyper-hydroxymethylation in MSCs from old subjects appeared associated with 5mC loss, thus suggesting a functional role of this modification in DNA methylation changes during aging [90]. Epigenetic changes are reversible; therefore, an epigenetic reprogramming is under study. Epigenome manipulation could be a way to promote self-renewal capabilities of MSCs, increasing their longevity and potency by contrasting senescence induced by culture expansion. Other studies are focusing on senolytic drugs to selectively remove senescent cells from MSC preparations [69,99].

The role of senescent MSCs in tumor progression is still elusive: cellular senescence can act on one side as tumor suppressor (by blocking expansion of damaged cells through an irreversible growth arrest) and on the other side as tumor promoter [80,100] through the SASP mediators released in the tumor microenvironment able to stimulate cancer cell migration and proliferation [67,101]. There are some very interesting reports demonstrating that in vitro aged MSCs can shift their activity and can promote tumor cell proliferation and migration. Therefore, even if MSC do not become transformed per se, they can negatively influence resident tumor cells. In a recent paper, it was shown that the conditioned medium (CM) from replicative senescent ASCs can promote tumor cell proliferation, possibly by induction of galectin-3, a protein strictly related to carcinoma cell proliferation, transformation and invasiveness [101]. In another paper, it was shown that umbilical cord senescent MSCs (induced to senescence by replicative or oxidative stress) markedly promote proliferation and migration of breast cancer cells, with the involvement of the Interleukin-6/STAT-3 (signal transducer and activator of transcription 3) pathway. The promoting effect was confirmed in vivo in mice by co-injection of tumor cells and senescent MSCs [80].

It is clear that in vitro expansion should be performed with caution and expansion times should be reduced as much as possible. Senescence and DNA damage acquisition in culture should be carefully determined before clinical application and on a case-by-case basis, especially in patients with a history of cancer. MSC potential to promote tumorigenesis is worthy of concern for clinical applications. This highlights the risks related to long-term cultures [69,101]. In addition to a possible relationship to transformation, the senescence can have a detrimental effect on cellular therapy efficacy, by affecting the biological properties of the cells. It appears therefore of critical importance to monitor senescence by defining reliable molecular markers of cellular aging and to asses MSC senescent status before clinical use. In general, for clinical safety issues and in an attempt to minimize the administration of senescent cells, the expansion should be reduced as much as possible (according to Torre et al. [102] it should not exceed four passages). Different biological processes and molecules have been proposed to evaluate cell senescence: proliferation arrest, telomere attrition, increase in senescent markers such as β-Galactosidase, DNA damage accumulation, increase in some proteins involved in cell cycle regulation such as p16 and p21, DNA methylation, prelamin A, SASP, microvesicles (MVs) [58,82,83,92,103,104]. Microvesicles dynamically change in relationship to the MSC status. Lei et al. [104] demonstrated that microvesicles from senescent MSCs resemble their parental cells; in particular, microRNA miR-146a-5p appeared up-regulated and most of its target genes were down-regulated in both MSCs and MSC-MVs during senescence, suggesting miR-146a-5p as a potential senescent marker to identify and monitor senescent MSCs.

## 5. Tumorigenicity

Tumorigenicity is one of the most relevant risk factors to be considered for MSC clinical applications [105]. Malignant transformation is a complex and progressive process involving several steps in which cells sequentially accumulate genetic alterations and modify their expression profile. In an interesting study, the transcriptional dynamics of immortalized human MSCs were analyzed by whole transcriptome analysis in association with the progression of cell transformation and chromosomal changes. The described series of genetic alterations with the parallel gene expression modifications could be helpful to establish key steps to assess transformation risks in MSC cultures [106]. Potential tumorigenicity of MSCs, and in particular of in vitro expanded MSCs, is a critical issue and it is strictly related to genomic instability. The absence of transformation potential must be demonstrated before clinical use. Unfortunately, cancer development is a long process and long follow-up of treated patients would be needed to verify safety in this context.

Critical aspects include MSC transformation potential in vitro; MSC ability to form tumors in vivo and MSC promoting effect on tumor development.

In the early 2000s, some impressive papers described MSC in vitro spontaneous transformation, but they were retracted following the demonstration that tumor cells were cancer cell lines cross-contaminating MSC cultures [106,107,108,109]. This highlighted for the first time the need to demonstrate the identity of cell preparations. At present, only one paper described spontaneous tumorigenic transformation associated with genomic alterations in culture. Long-term cultivation (beyond five weeks) of bone marrow- and liver-derived MSC produced transformed cells (four batches out of 46) able to induce sarcoma-like tumors in immunodeficient mice. High-resolution genome-wide DNA array and short tandem repeat profiling confirmed a shared origin of the transformed cells and parental MSC [110]. By gene and microRNA expression arrays authors also identified a gene expression signature that may potentially serve to screen cultures for evidence of early transformation events [110]. The results of this publication have not yet been confirmed. MSC spontaneous transformation in vitro appears a very rare event eventually occurring after a relatively long-term culture. Overall, for the expansion protocols commonly used in clinical MSC preparations, malignant transformation appears not to constitute a substantial safety issue as long as early MSC passages are used.

In contrast to iPSC and ESC showing higher instability and potential to form teratomas [64,111,112], MSCs do not appear intrinsically tumorigenic [7,71,74,105,113,114,115,116,117,118,119,120,121,122,123,124]. The epigenetic repression of the pluripotency factor OCT4 in adult stem cells can partly explain this lower tumorigenic potential [78]. Several papers demonstrated the absence of tumorigenic potential of cultured MSCs of different tissue origin even at advanced in vitro culture times [48,125,126,127]. Accordingly, long-term in vivo tumorigenic assessment of in vitro expanded ASCs in nude mice evidenced that one year after injection, transplanted cells were completely removed and no tumors were present [128]. In another study, subcutaneously injected ASCs in immunodeficient mice were still present after 17 months without signs of migration from the injection site and without teratoma formation [117]. In humans, since the first clinical trial using MSCs in 1995, tumors have never been reported [129,130]. However, eight years after intraspinal olfactory mucosal cell autoimplantation for spinal cord injury, a young patient developed a spinal cord tumor mass autograft-derived. The mass histologically resembled olfactory mucosa and produced mucus [131]. Moreover, a case of glioneuronal brain tumor developed four years after fetal neural stem cell transplantation to treat ataxia telangiectasia was described in 2009. By genetic typing, it was demonstrated that tumor cells were of donor origin [132]. In the first case, autologous adult stem cell grafts were used, and the tumor developed many years after transplantation. Autologous treatments appear more dangerous by this point of view since they are likely to be less immunogenic and more long-lasting than allogeneic ones. These cases highlight the need to perform long follow-up studies.

If on one side it seems that MSCs do not pose an obvious risk of tumorigenesis having a low possibility to undergo oncogenic transformation, on the other side they can be tumor promoting by stimulating growth and dispersion of resident tumor cells already present in the MSC recipient. Actually, in vivo MSCs have been demonstrated to play a role both in tumor formation and suppression, in a complex interplay between cells and soluble factors. The controversial results available in the literature concerning the promoting or suppressive MSC activity can be mostly attributed to the activation state of MSCs. As for monocytes/macrophages, MSCs can be activated in two different phenotypes, a pro-inflammatory phenotype (MSC1) and an immunosuppressive phenotype (MSC2) [133]. In accordance, it was demonstrated that MSC1-based therapy attenuates tumor growth whereas MSC2 treatment promotes tumor growth and metastasis [134]. This highlights that unstimulated MSC or MSC2 (able to promote regeneration and therefore useful for regenerative applications) have also the potential to promote tumor growth, whereas MSC1 do not promote regeneration nor tumor growth.

Moreover, MSC ability to suppress immune responses may contribute to tumor growth promotion and metastasis of resident tumor cells [9,135,136]. MSC paracrine activity can promote tumor growth through several mechanisms: immunosuppression, promotion of angiogenesis, contribution to the tumor microenvironment, inhibition of apoptosis and promotion of metastasis [137]. In vivo injected MSCs sustained cancer growth in mice by suppressing the recipient antitumor response [138]. Tumor promotion was shown to be also mediated by the MSC secretome alone. Conditioned medium from healthy donor-derived BM-MSC was shown to promote tumor cell proliferation, glucose uptake and vascularization via up-regulation of c-myc in tumor cells [139]. Human BM-MSC-derived exosomes sustained tumor growth and angiogenesis in mice, at least partly by upregulating tumor cell expression of vascular endothelial growth factor (VEGF) through extracellular signal-regulated kinases 1/2 (ERK1/2) pathway [140]. In another recent study human BM-MSC-derived exosomes promoted in vitro tumor growth of osteosarcoma and gastric tumor cell lines through the activation of the Hedgehog signaling pathway in cancer cells [141,142]. Human ASCs were demonstrated to promote angiogenesis of breast tumors in vivo in a nude xenograft model through secretion of CXC chemokine ligand 1 and 8 (CXCL1 and CXCL8), thus underlying the risk of cancer recurrence after fat graft in reconstructive surgery for breast cancer patients [142]. On the opposite side, cancer-primed MSCs profoundly modify their secretome and partially lose their anti-tumor activity by abrogating the production of pro-senescent and apoptotic factors [67]. All these data suggest particular caution in applying MSC for regenerative approaches in recipients with coincidental tumors.

## 6. Genetic Stability

A high number of studies have well documented that human MSCs acquire cytogenetic alterations in culture both in autosomes and in sex chromosomes, even if they are frequently transient and not clonal [28,29,55,73,74,130,143,144,145,146,147,148,149,150,151,152,153]. Conversely, other studies reported stable chromosomes in culture for MSCs of different tissue origin [24,33,36,48,116,118,120,124,125,126,127,149,153,154,155,156,157,158]. Depending on the studies, tissue sources, culture conditions, culture times, these alterations occur at early or late passages and at different frequencies, thus explaining inconsistencies in literature data (for a detailed list of described aneuploidies, see Rebuzzini et al. [148]).

The relevance of the clear tendency towards increased DNA alterations with culture passages is still being debated [148,149,151,159,160] since genomic instability, particularly chromosomal alterations, is a hallmark of cancer [161]. Two risks can be envisaged: tumorigenicity on one side and impaired MSC biological activity on the other side. Tarte et al. [151] claimed that karyotype and Fluorescent In Situ Hybridization (FISH) are neither adequate nor informative as release criteria since cells with or without aneuploidies became senescent without evidence of transformation either in vitro or in vivo [74,119]. Even if chromosomal abnormalities can arise in vitro, the entire transformation process is not reached in culture [76] and oncogenic transformation does not necessarily associate with clonal expansion and growth advantage in vitro [95]. In addition, in vitro oncogenic transformation of human cells can proceed without widespread genomic instability [162].

However, a 4% incidence of aneuploidies was described in a large set of MSC preparations by Ben David et al. [143], with a correlation between MSC aberrations arising in culture and the most common aberrations found in tumors of the corresponding tissue. The relevance of these observations was argued by Sensebé et al. [149] priming a debate [160] that is still open [159] between supporters of the irrelevance and transient nature of these alterations [119,149,151], and detractors [105,143], that suggest more caution in result interpretation. Actually, some alterations appear advantageous and are selected in culture [143,146] thus representing a potential risk, even if in most cases altered cells did not overcome senescence and did not immortalize [146]. Moreover, cytogenetic alterations were observed during in vitro expansion of ASCs and umbilical cord-derived MSCs in heterogeneous subpopulations, with sequential dynamic changes similar to those described for tumor cell populations during disease progression. Aberrant clones arising at earlier passages become predominant at later ones [75].

A conventional karyotype analysis performed on a large set of clinical-grade BM-MSC showed only spontaneous nonclonal aneuploidies, therefore authors assumed that absence of clonal anomalies and a cut-off of 10% nonclonals can be chosen as release criterion [28,144]. Roselli et al. [28] evaluated the genomic stability of chorionic villi-derived MSC by karyotype, array comparative genomic hybridization (CGH) and microsatellite instability analysis (MSI). They found abnormal clones only beyond P10 and these clones showed neither growth advantage nor senescence resistance in accordance with Tarte et al. [151] that reported aneuploidies in culture without transformation signs. Some authors suggest that if cells become senescent in culture, they are unlikely to produce tumors in patients, even if actually there are no methods available that allow us to exclude the presence of few tumorigenic cells harboring oncogenic mutations [163]. Microsatellites appear stable thus indicating that the Mismatch Repair System (MMR) efficiency is maintained in MSC cultures [28,55,81].

Kim et al. [75] performed karyotype and FISH analysis of cytogenetic aberrations on a large case series of MSC preparations (68 cell preparations of different tissue origin) to establish reference values for aneuploidy in human MSCs. They found variable aneuploid clonal proportions (1%–20%), with asymmetric aneuploid patterns and with chromosomes 16, 17, 18 and X mostly involved [75]. They suggested an aneuploidy cutoff of 13% as well as to combine chromosome. 16, 17, 18 and X FISH analyses to assess MSC genetic stability.

Binato et al. [164], reported changes in the expression of key genes from passage 5 on and therefore suggested to use cells up to passage 4 and to analyze case by case cells from higher culture passages.

Even if there is no definitive evidence of in vitro MSC transformation and the majority of the reported anomalies leads to senescence, it is fundamental to perform an accurate analysis of genetic stability before clinical application. Moreover, genetic alterations have also the potential to affect the therapeutic efficacy of the cells. Therefore, it appears of critical importance to know how many passages can be performed before cells acquire alterations and lose their therapeutic properties. With this in mind, cellular models should be forced to senescence by long-term culture and genetic alterations should be verified at different time points to establish the optimal culture times [165]. Donor characteristics, the tissue of origin and culture conditions are important factors to be considered.

### Methodological Approaches to Assess Genetic Stability

Available guidelines do not identify exclusive tests to assess genetic stability. Therefore, how to assess genetic integrity of in vitro expanded cells? Several methods are available today to identify/quantify genetic/epigenetic alterations, with different targets, different resolution, different execution times and costs. The vast majority of techniques fails in detecting alterations affecting less than 10% of the analyzed cell population and this represents an important issue for the early detection of newly occurring somatic mutations. Moreover, all genetic variants detected need a biological interpretation.

In most cases standard karyotype is performed, that is a very useful and consolidated technique, but with significant drawbacks, first of all the low sensitivity: it fails in detecting rare cell populations (metaphase cells can represent as few as 0.01% of the tested cells) and it shows a resolution in the order of megabases of DNA. Moreover, it is time-consuming, it is hampered by the difficulty of obtaining sufficient metaphases and it requires highly qualified personnel. In a research setting, more sensitive techniques like array-CGH are also used, but we are still in the order of hundreds of kilobases (kb). Interphase FISH is also frequently used in association with standard karyotype. Moreover, DNA sequence and expression levels of key genes involved in cell cycle regulation, senescence and cancer are frequently evaluated using different techniques. In the research setting, other techniques have been evaluated to check for universal biomarkers, e.g., micronuclei and nuclear blebs quantification [166,167] and virtual karyotyping [168], the latter allowing for the evaluation of genomic integrity of autosomes based on the comparison of gene expression profiles of the sample of interest with a reliable database of the same cell type and the same microarray platform. In Table 1, the most frequently used techniques for genetic stability assessments with principal pros and cons are listed.

Cell transformation is associated not only to chromosomal anomalies such as gross deletions, duplications and aneuploidies but also to micro-deletions and -duplications and mutations at the single base level.

It seems appropriate to extend the analysis of genetic alterations acquired in culture to small molecular defects since there is no reason to assume that small genetic changes are irrelevant. At present, a full DNA characterization of MSCs through comprehensive studies on small molecular genetic and epigenetic alterations is still lacking, but it would be extremely helpful in detecting possible frequent mutations in order to increase knowledge about MSC behavior and to set up increasingly targeted biosafety tests.

The availability of high throughput technologies such as next-generation sequencing (NGS) makes it possible today to obtain this information of critical relevance. NGS appears an ideal candidate for this purpose thanks to its very high and scalable throughput, sensitivity and accuracy with affordable costs and a clear gain in efficacy and timing, with the potential to be translated to the clinical practice. NGS massively parallelizes genomic interrogation thus allowing for detection of very rare alterations (i.e., for early detection of the occurrence of adverse events). To confirm NGS data an equally or more sensitive technique is required, therefore Sanger sequencing (with a detection rate of about 10% of the analyzed cells) is unsuitable. Droplet digital PCR or pyrosequencing could be used.

Given NGS versatility, different approaches can be used: whole exome sequencing (WES) or whole genome sequencing (WGS) allowing for a complete analysis of the genome, but with a limited read depth; or targeted deep sequencing of specific genes or DNA regions of interest, with the possibility to reach very high coverage depth and to detect very rare alterations.

Some recently described NGS applications were useful to understand the contribution of BM-MSC molecular alterations in the pathogenesis of acute myeloid leukemia (AML). Whole exome sequencing (WES) of AML BM-MSC from 21 patients revealed a non-specific mutation pattern compared to BM-MSC from healthy donors showing no alterations. This strongly suggests global changes in MSCs possibly altering the tumor microenvironment and influencing tumor behavior [169]. Another study on the same topic used NGS technology to focus, by targeted resequencing, on a panel of 50 genes involved in myeloid malignancies. They found neoplastic cell-specific mutations as well as genetic variants shared with BM cells and MSC specific genetic variants, the last at a very low-frequency [170]. Kim et al. [71] analyzed by NGS a series of cancer related genes during in vitro culture of human umbilical cord blood MSCs and found mutations occurring at P16 in NOTCH1 (NOTCH homolog 1), MLH1 (MutL homolog 1), GNAS (guanine nucleotide binding protein, alpha-stimulating activity) and TP53 (tumor protein 53) genes. Heterogeneity among clones with distinct chromosomal aberrations dynamically changed over time, similar to what is observed in cancer stem cells. Cai et al. [59] analyzed by WGS genetic dynamics in culture (from P1 to P13) of one BM-MSC sample. No significant copy number alterations and low levels of single nucleotide changes (SNCs) were observed until P8, whereas numerous SNCs were observed at P13. Very interestingly, the genetic alterations did exist in uncultured cells with a low allelic frequency (undetectable with traditional methods) but reached up to 36% in passage 13. Results argue for a relatively stable genome in the early passages, but with a significantly increased occurrence of altered clones at advanced passages, thus suggesting dominant clonal growth and highlighting the relevance of the alterations already present in the MSC donor. We also performed a preliminary study to detect somatic variants in cancer-related genes in human ASCs expanded in vitro and identified some variants of predicted pathologic significance not present at seeding and occurring in culture (unpublished preliminary data).

Progressive accumulation of comprehensive studies in a large series of samples will potentially allow to identify recurrent mutations specific for certain cell types and to set up defined panels of alterations to be checked in a clinical setting before cell administration. The critical point will be to demonstrate the biological relevance of the observed alterations and the real risk they represent [111]. Long-term follow-up will be of help in understanding this point.

Another critical point to be considered is related to patient characteristics. Besides the occurrence of genetic alterations in culture, there is also the risk of expanding cells with DNA alterations already present in vivo, as demonstrated by papers that detected mutations already present at cell seeding, even if at a very low-frequency [59]. Moreover, even if without starting alterations, donor cells can possess an intrinsic propensity to genetic instability that will be developed in vitro [55,150,151]. This poses an important question about the eligibility of MSC donors, possibly harboring deleterious mutations due to genetic or acquired pathologies in their MSC reservoir, with the potential to be expanded in vitro. For example, MSCs from myelodysplastic syndrome patients were demonstrated to be genetically unstable by karyotyping and FISH. Some alterations occurred at the first analyzed culture passage (P2). Therefore, it cannot be established if they were already present in vivo. Interestingly, there was no overlap between MSC cytogenetic alterations and those of the hematopoietic compartment [171]. BM-MSC from Fanconi Anemia patients cultured in vitro until P5 display chromosomal fragility (but not unbalanced rearrangements) and signs of senescence occurring earlier compared to BM-MSC from healthy subjects [172]. Conversely, in another study, autologous ASCs from oncologic patients did not show tumor-associated genomic alterations but only transient variations [130]. Accordingly, Lucarelli et al. [147] did not detect significant differences in genetic stability (by karyotyping, array-CGH and sequencing of cancer-related genes) of BM-MSC from sarcoma patients compared to controls. The influence of donor characteristics on the genetic stability of MSCs was also observed in ASC from lipoaspirates where only one of the samples showed in vitro early deceleration of growth and occurrence of autosomal aneuploidies, suggesting a particular susceptibility of the cells of this donor [55]. Similarly, a peculiar occurrence of different clonal cytogenetic alterations in different parallel cultivations of BM-MSCs from a 67-year-old patient has been described [173]. Cells were expanded for refractory ischemic cardiomyopathy treatment, but the patient had a previous history of kidney tumor. Despite the absence of cytogenetic alterations in a bone marrow sample of the same donor, cells displayed alterations from the second culture passage onwards, suggesting a donor-specific propensity to genetic instability [173]. All these data point to the need for a case by case assessment of genetic stability and to the need of a careful check of the health status and concomitant morbidity of MSC donors.

Finally, culture conditions can greatly influence the biological properties of MSCs [50,118,154,158,174] including genomic stability [148]. Supplements seem to not influence genomic stability [36], whereas oxygen tension is critical [175] exerting a marked regulatory effect on cell cycle checkpoint genes [166]. It was demonstrated that long-term cultures under hypoxic conditions prevent senescence [82,83,84] while maintaining MSC differentiation potential. BM-MSCs cultured under hypoxia improved their DNA damage response and DNA repair (both non-homologous end-joining, NHEJ, and homologous repair, HR) [84]. This effect was not observed in ASCs, in accordance with different in vitro characteristics displayed by MSCs depending on the tissue of origin [176]. Another aspect to be evaluated is cryopreservation. This procedure is required to pool cells for off-the-shelf clinical applications but raises safety issues in terms of possible cell alterations or cell selection during freezing-thawing passages. To date, literature data point to the stability of cryopreserved MSCs [28,174,177]. Studies on the genetic stability of MSCs should also be performed in genetically engineered cells after modifications possibly affecting their genome and to evaluate genotoxicity, such as genetic modifications using vectors, transfer of nanocomplexes and nanoparticles. Recently, a specialized flow-cytometry-based method was developed to quantitatively analyze genotoxicity while determining at the same time the mode of mutagenic activity [178]. It was shown that highly positively charged lipid- and polymeric-based vectors can induce genotoxicity, thus highlighting the need for extreme care in evaluating the safety of ex vivo modified MSCs [178]. Functional nanoparticles used as therapeutics and transferred to human MSCs did not affect genome integrity or increased the mutation rate as evaluated at chromosome and single base level [179].

## 7. Regulatory Aspects

The increase of therapeutic interventions based on MSCs created an urgent need for specific legislation for the approval and monitoring of their clinical use. Actually, many dangerous unregulated stem cell treatments are ongoing across the world [180] and a great effort is in progress to efficiently regulate this field. To date, no stem-cell medicinal products have received marketing authorization in Europe. Access to stem-cell medicinal products is allowed under certain conditions, e.g., taking part in clinical trials or compassionate-use programs, or receiving a custom-made medicine (European Medicines Agency, EMA/763463/2009).

Institutional review and ethics committees need specific rules to approve and evaluate preclinical and biosafety demonstration tests on cellular products. Since the scientific field is still in progress, guidance is subjected to constant update to continuously translate accumulating scientific knowledge to therapeutic strategies in order to reduce the risks. Stem cell use requires authorization from national regulatory agencies from the countries involved in the clinical trials. From a regulatory point of view, there is the need for the development of diagnostic tools to definitively recognize clinically “safe” and “unsafe” cell products. Controls should guarantee microbiological safety and the absence of potential side effects linked to genomic instability driving transformation, reduced potency or senescence and must be standardized [181]. Moreover, standardized guidelines for isolation, expansion, preservation and delivery of MSCs are needed to minimize variability and to allow for comparison of different studies. Quality issues to be demonstrated include purity, tumorigenicity, potency to measure biological activity (with surrogate markers), biodistribution.

In Europe, MSC preparations are considered as advanced therapy medicinal products (ATMPs) when their use is preceded by substantial manipulation (including in vitro expansion) or if they are used for a different essential function. They can be somatic-cell therapy products or tissue-engineered products (engineered cells or tissues administered to human beings to regenerate, repair or replace a human tissue). In these cases MSC use is subjected to ATMP specific regulation (available online: https://www.ema.europa.eu/en/human-regulatory/overview/advanced-therapies/legal-framework-advanced-therapies). Reference legislation is summarized in Table 2 (see also References [37,182]). In addition, the EMA produces scientific recommendations and guidelines on ATMP use, drawn up by specific Committees (Table 2). The Committee for Advanced Therapies (CAT) is responsible for classification, certification and assessment of quality, safety and efficacy of ATMPs; the Committee for Medicinal Products for Human Use (CHMP) is involved in the development facilitation of ATMP-based therapies [183].

The 2003/94/EC directive establishes quality control procedures ensuring product standardization and traceability and also the management of abnormalities. Criteria for quality and safety must be in accordance with 2004/23/EC and 2006/17/EC. After structure accreditation in accordance with 2001/83/EC and 1394/2007, the CAT is responsible for documentation evaluation and conformity to 2009/120/EC [37].

ATMPs must meet the same stringent requirements of drugs requiring production processes in line with Good Manufacturing Practices (GMP) and the guarantee of product safety by controls not only at batch release but also in all production phases [102,181]. GMP compliance must guarantee the delivery of safe and reliable products, therefore most of the limitations do not apply to the products but rather to the processes. ATMP development is not based on fixed pre-established criteria, but it should be adaptable case-by-case in order to face with the complexity and heterogeneity of cell therapies, that is why reliable quality control of MSCs is yet elusive [184]. Clinical grade stem cell products and procedures must be used also in preclinical research to allow predictable translation to the clinic [105]. Preliminary safety data must be provided to authorities and to the CAT; activity, safety, efficacy and required dose must be defined before clinical use. What is strongly recommended is a risk-based approach covering the entire development of ATMPs. In particular, article 3.3.2.3 (regulation 2009/120/EC) identifies which relevant information is needed: identity, purity (e.g., adventitious microbial agents and cellular contaminants), viability, potency, karyology, tumorigenicity and suitability for the intended medical use. The same article also reports that “genetic stability of the cells shall be demonstrated”, but no indication about technical procedures to be used is present. In fact, accurate guidelines and standardized tests for the evaluation of genomic instability have not currently been established. Conventional karyotyping combined with other techniques like comparative genomic hybridization (CGH) and fluorescent in situ hybridization (FISH) is agreed as the state of the art methodology to evaluate possible chromosomal aberrations. These analyses should be performed in each batch only when recurrent alterations (present in two different cultures from the same donor) are detected [165]. The CAT suggests performing cytogenetic analysis, telomerase activity, proliferative capacity and senescence. However, these analyses are not mandatory and no defined protocols are recommended.

A provocative debate of the scientific community was triggered by impressive papers reporting in vitro malignant transformation of MSCs, although followed by retraction [107,109]. Therefore, in 2011 the Cell Products Working Party (CPWP) arranged an expert meeting to discuss the challenges of MSC-based therapies, focusing on tumorigenicity [165]. The need to perform cell culture under GMP conditions to ensure cell segregation and strict controls was emphasized. Since cell propagation and culture conditions (such as presence/absence of serum) significantly influence the occurrence of cytogenetic anomalies [122], it was suggested to perform a reduced number of population doublings in slow growth conditions. A karyotype or FISH at batch release is required only if recurrent chromosomal anomalies were found [165].

In general, no detailed requirements are indicated, therefore applicants must develop a risk assessment program for their specific product. A tentative guideline with minimal MSC quality requirements for clinical use was proposed by the Italian Group of Mesenchymal Stem Cells (GISM) with indications for standardization and optimization of critical points such as cell isolation, in vitro expansion, validation (characterization, functionality, potency and safety) and quality control for identity, sterility, tumorigenicity and genomic stability. In particular, to exclude potential tumorigenicity, assessment of telomerase activity (that should be low/undetectable) and soft agar test are recommended. For genomic stability, karyotyping is recommended. The quality control process is foreseen during the validation phase, during production and at batch release [102].

For tumorigenicity testing, an approach combining in vitro and in vivo studies or a combination of in vitro studies is suggested, without specification of preferred tests [183]. The proposed in vitro assessments are growth rate, anchorage-independent growth by soft agar culture, cytogenetics, cell differentiation, functionality, expression of cell-cycle regulation genes, oncogenes and tumor-suppressor genes, telomerase activity and senescence. However, all these indirect tests do not guarantee the absence of tumor formation in vivo. In vivo studies are required anytime substantial manipulation is performed and generally foresee the use of immunodeficient animals with the same route of administration and a follow-up depending on biodistribution and persistence studies [165,183]. Anyway, a tumor developing in vivo on an immunodeficient animal model will not necessarily develop in vivo in humans. Preclinical nonxenogeneic studies using animal transplant models before the development of human equivalents have been suggested as the actual most relevant method to assess tumorigenicity [105]. It should be noted that the frequency of MSC transformation could be too low to be detected in rodent models and the immunological status of the animal can influence the results. Before any definitive statement on MSC tumorigenicity can be made, the risk of tumor formation must be monitored in the clinical setting with longer follow-up than those available today [165] taking into consideration that tumors need many years to develop [131,132].

The International Society for Stem Cell Therapy (ISSCR) recently revised and extended the past guidelines for stem cell research and clinical translation (ISSCR, 2006 and ISSCR, 2008) accounting for scientific progress (such as mitochondrial replacement and genome editing), policy development and new ethical concerns and social priorities. These guidelines provide principles and best practices for basic, translational and clinical research [185] addressing ethical and scientific issues. One of the points is that release criteria must be designed to minimize risk from culture-acquired abnormalities. Preclinical trials must characterize cells, evaluate potential toxicity, possible tumorigenicity risks, biodistribution and long-term effects. Moreover, a detailed biodistribution study is mandatory. Practices to address long-term risks and to detect unforeseen safety issues must be adopted. Guidelines also provide indications about the use of small and large size animal models. Guidelines recommend rigorous demonstration of safety and efficacy in preclinical studies and that clinical trials be subject to rigorous peer review focused at evaluating the risk/benefit balance. Research must be reviewed and approved. Both positive and negative results must be published. In particular, risk of tumorigenicity must be rigorously assessed for any stem cell-based product, especially if extensively manipulated in culture, genetically modified or when pluripotent. An acceptable balance of risk and clinical benefit must be demonstrated. Particular attention should be paid to research integrity and fairness, patient protection, welfare and social justice, transparency, the centrality of good clinical practice, solid preclinical data and review processes, improved informed consent, rigor in research, public responsible communication avoiding to exaggerate potential benefits and underestimate risks [85].

To sustain ethic and institution review boards in the approval process of cell-based trials, the ISSCR developed a framework of questions covering the principal issues including: mechanism of action, mode of delivery, presence of preclinical safety and efficacy demonstration, GMP production, reagent quality control, potency assays, Phase I trial requirements [41,186]. By using this questionnaire stakeholders will be helped in ascertaining if there is sufficient scientific and preclinical support for moving to a clinical trial thus protecting patients and allowing translation of research to the clinic.

## 8. Concluding Remarks

Research data clearly demonstrate an intimate correlation between replicative potential, stemness, senescence and genetic stability of human MSCs, but comprehensive studies and standardization of procedures are still required to completely explore MSC biology. Basic research on MSCs is fundamental to evaluate all the still open issues about these cells and the extremely powerful technical tools now available have the potential to help scientists in reaching this goal. This will help in supporting clinical trials by strong scientific rationale and adequate preclinical safety and efficacy testing to avoid not only safety concerns but also excessively high restrictions to MSC-based promising therapies.

Currently, the available data point to the need of a case by case evaluation of MSC clinical applications, particularly in relation to donor characteristics, specific culture conditions and specific therapeutic applications. Genetic stability and senescence appear critical aspects to be carefully evaluated to avoid failures or safety risks and long follow-ups are needed to monitor the incidence of possible side effects at longer times.

Interpretation of genetic instability and senescence of cultured MSCs is controversial, but the increasing incidence of genetic alterations at advanced culture times clearly indicates that few culture passages correspond to a reduced chance to harbor dangerous alterations. Therefore, a prudential behavior is desirable with reduction of culture times as much as possible to avoid safety concerns.

Finally, an accurate evaluation of the balance between risks and benefits appears a fundamental criterion to guide the choice to use or not a therapy, particularly for treatment of non-life threatening pathologies where the risk/benefit ratio has to be carefully estimated.

## Figures and Tables

**Figure 1 ijms-20-02406-f001:**
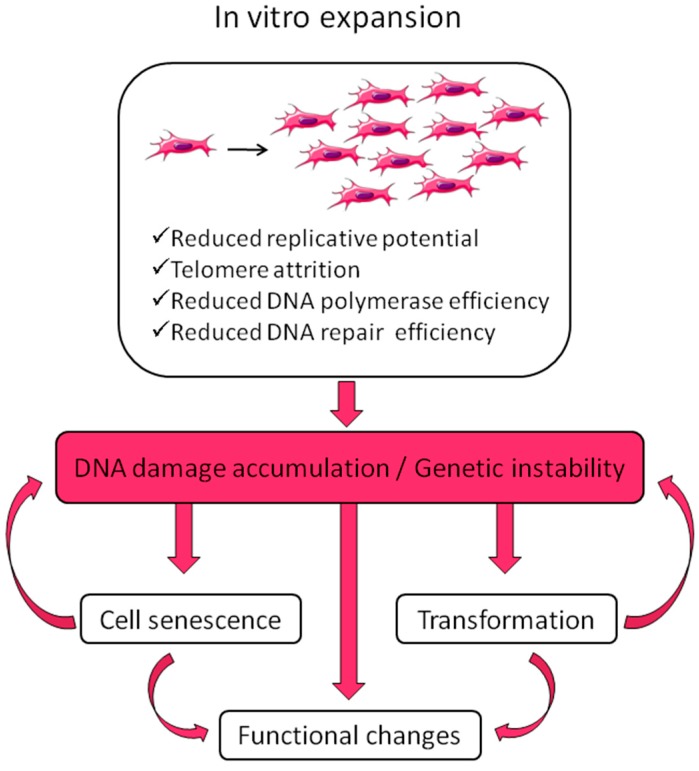
Effects of in vitro expansion. During in vitro expansion cells reduce their replicative potential and accumulate DNA damage due to progressively reduced DNA synthesis and repair efficiencies. DNA damage accumulation can affect genomic integrity of the cells possibly driving senescence and transformation, with consequent functional alterations. These can impair the therapeutic effect and raise safety issues.

**Table 1 ijms-20-02406-t001:** Techniques to assess genetic damage. Methods are listed based on their level of resolution.

Method		Detected Alterations	Resolution	Target	Characteristics and Limits
Conventional Karyotype (G, Q banding)	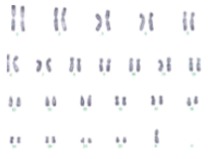	Chromosomal rearrangements, aneuploidies, deletions, duplications	5–10 Mb	Whole genome	Low throughput Only gross alterations. Interphase cells and low-frequency alterations not detected.
SKY (Spectral Karyotype)	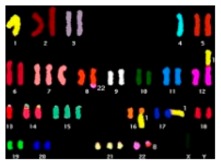	Structural abnormalities (small and complex rearrangements), aneuploidies	1–2 Mb	Whole genome	Inversions, deletions and duplications in the same chromosome not detected. Interphase cells and low-frequency alterations not detected.
Array-CGH (comparative genomic hybridization)	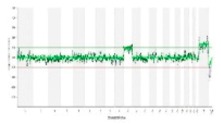	Deletions duplications	≤50 kb	Whole genome	Low mosaicism (20%–30%) and balanced rearrangements not detected.
Virtual Karyotype (e-karyotype)	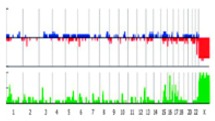	Deletions, duplications	20kb–1 Mb	Whole genome	Comparative large scale expression analysis. Can compare different dataset.
FISH Fluorescent in situ hybridization	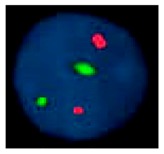	Structural abnormalities/aneuploidies	3–10 kb	Specific targets identified by probes	Can be performed in interphase cells.
Microsatellite instability (MSI) analysis	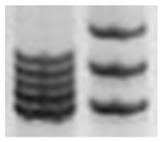	Repeated sequences variations	2–6 bp	Repeated sequences throughout the genome	Indirect indication of genomic instability
Single nucleotide polymorphism (SNP) array	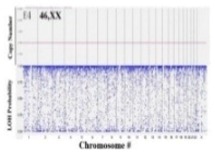	Single base modifications	1 bp	SNP throughout the genome	Detects single base variations in potentially hundreds of thousands of loci
gammaH2AX	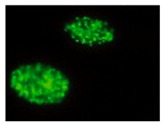	Histone phosphorylation		Whole nuclei	Indirect measure of double-strand breaks
Telomere length by Southern	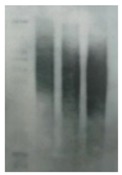	Telomere attrition	0.5–10 kb	All telomeres	Labor intensive and time-consuming.
Sanger sequencing	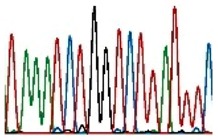	Single base modifications/small deletions and duplications	1 bp	Specific regions identified by primers	Do not detect low-frequency alterations.
Next generation sequencing (NGS)	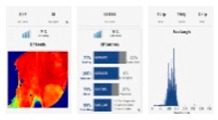	Single base modifications/small deletions and duplications	1 bp	Whole genome, exome, specific regions	High throughput, scalable.
Micronuclei test	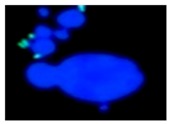	Micronuclei and blebs		Whole nuclei	Macroscopic alterations, labor intensive and time-consuming, only semi-quantitative
Comet assay	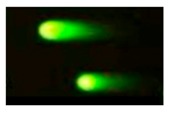	Aspecific DNA fragmentation		Whole cells	Aspecific indication of DNA damage
Droplet digital PCR	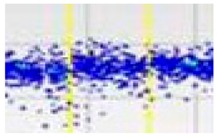	Single base variations	1 bp	Specific regions identified by primers	Very high sensitivity (0.01%)

**Table 2 ijms-20-02406-t002:** The European regulatory framework on Advanced Therapy Medicinal Products (ATMPs).

**European Reference Legislation**	
Regulation (EC) N° 1394/2007 implemented by regulation EC N° 668/2009	Definition of ATMPs. Legal basis for authorization procedure of ATMPs.
Commission directive 2009/120/EC (amending directive 2001/83/EC)	Updated definition and detailed scientific technical requirements for gene-therapy and somatic cell therapy medicinal products combined ATMPs and tissue engineered products.
Commission directive 2004/23/EC implemented by directives 2006/17/EC and 2006/86/EC	Definition of quality and safety standards for donation, procurement and testing of human tissues and cells.
Regulation EU 536/2014 repealing directive 2001/20/EC	Implementation of good clinical practice in the conduct of clinical trials on medicinal products for human use.
**Guidelines and Recommendations**	
EMEA/CHMP/410869/2006	Guideline on human cell-based medicinal products adopted in 2008. General overview of the requirements to license ATMPs.
EMEA/149995/2008	Guideline on safety and efficacy follow-up and risk management of ATMPs.
EMA/630043/2008	Update of the procedure for the evaluation of ATMPs marketing authorization, adopted in 2018.
EMA/CAT/571134/2009	Reflection paper on stem cell-based medicinal products adopted in 2011. Focused on stem cell-based products.
EMA/763463/2009	Public statement on concerns over unregulated medicinal products containing stem cells.
EMA/CAT/600280/2010 rev.1	Revision of the reflection paper on the classification of ATMPs. Major points: what constitutes a substantial manipulation of cells or tissues; what is considered as a non-homologous use of cells or tissues.
EMA/CAT/CPWP/686637/2011	Guideline on the risk-based approach for ATMPs’, adopted in 2013.
C (2017) 7694 Guidelines	Guideline on GMP specific for ATMPs. Recommendations: risks and effectiveness based on current scientific knowledge; level of effort and documentation commensurate with the risk.
EMA/CAT/327664/2018	CAT work plan 2019 including the development of a new guideline on requirements for ATMPs in clinical trials and on ATMP comparability.
In preparation	EMA guideline on investigational ATMPs to create common standards for the assessment of novel ATMP products.Public consultation for the draft revised guideline EMEA/149995/2008) closed. Outcome expected in 2019.

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
