# Peer review of "Genetic Stability of Mesenchymal Stromal Cells for Regenerative Medicine Applications: A Fundamental Biosafety Aspect"

_ijms, 2019, doi:10.3390/ijms20102406_

Round 1

Reviewer 1 Report

The Review discusses the genetic stability of mesenchymal stem cells for regenerative applications and from a safety standpoint. The manuscript is well written, and the topic is relevant and actual. Minor corrections will further improve the quality of the message in the text.

1.       In particular, it would be important to better highlight if specific epigenetic factors (which exogenous factors can really influence genetic stability and safety of MSC?)  have been associated to genetic alterations due to MSC injection. Some examples should be provided. For instance, pharmacological regimens accompanying MSC-base therapies could influence this aspect? The origin of the potential instability of MSC is also related to donor’s response?

2.       Line 77 and line 96 authors should only report the reference number omitting the sentence reviewed by…

3.       I would reduce the general introduction by referring to MSC and I would have combined with the second paragraph on the clinical use. Most reviews on MSC point out the typical features of these progenitors, so I would avoid further repetitions, directly focusing on the genetic side of MSC.

4.       The genetic alterations of MSC should be better described: what they consist of? Author should be a little bit less general (i.e. DNA proliferation and/or alteration). It is important not only to provide information but mainly to challenge the reader about a specific hypothesis. What does the author reckon regarding the genetic instability? What can determine this long term effect in MSC? It is only a matter of in vitro manipulation of the cells or the interaction with the microenvironment that it should be taken into account? The genetic instability has been also described for the homologue use of MSC?

5.       The sole observation of cytogenetic aberrations in culture of MSC can necessarily imply a tumorigenic effect in vivo? The genetic instability only represents a safety issue or it causes clinical failure? Additional non tumorigenic effects such as alteration of the immune response or lacking of differentiation capacity of MSC can occur? Author has separately discussed senescence and tumorigenesis. Perhaps, these two paragraphs could be merged by clarifying this aspect    

6.       Besides, standardized methodologies to assess genetic instability of MSC are currently employed in regenerative applications as readout of the clinical outcome? Or they only represent an ancillary quality control? Some are described at page 9.

Author Response

May 8, 2019 

Dear reviewer,

Thank you for your valuable comments and suggestions. I am very happy for your positive evaluation. Please, find below the detailed responses to your questions. Changes made and highlighted in the manuscript are also indicated.

The Review discusses the genetic stability of mesenchymal stem cells for regenerative applications and from a safety standpoint. The manuscript is well written, and the topic is relevant and actual. Minor corrections will further improve the quality of the message in the text

1.In particular, it would be important to better highlight if specific epigenetic factors (which exogenous factors can really influence genetic stability and safety of MSC?) have been associated to genetic alterations due to MSC injection. Some examples should be provided. For instance, pharmacological regimens accompanying MSC-base therapies could influence this aspect? The origin of the potential instability of MSC is also related to donor’s response?

MSC genetic alterations can occur either in the donor or in culture. They can be already present in cells isolated from the donor, due to donor characteristics such as age, pathologies associated to DNA instability, exposure to mutant agents like radiations or chemicals, cell aging. These alterations (mutations, deletions, duplications, epigenetic changes) are generally present at a very low frequency in vivo in the donor, but they can be selected during in vitro expansion (both positive and negative selection can occur) and therefore they can be expanded or eliminated, respectively (this is discussed in paragraph n.6). Moreover, in vitro replication per se is known to induce DNA damage accumulation. The present review focuses on genetic alterations expanded or acquired in culture, therefore before cell injection. In general, MSC genetic alterations arise before and not after cell administration, and MSC genetic instability is not attributable to the recipient (homologous or heterologous) or to the recipient microenvironment. I realize that probably this was not clearly explained and I added a sentence in paragraph 3 (page 3, lines 139-141).

2.Line 77 and line 96 authors should only report the reference number omitting the sentence reviewed by…

The text has been modified according to the suggestion.

3.I would reduce the general introduction by referring to MSC and I would have combined with the second paragraph on the clinical use. Most reviews on MSC point out the typical features of these progenitors, so I would avoid further repetitions, directly focusing on the genetic side of MSC.

I agree with the reviewer that some general information is frequently repeated in the introduction section of papers/reviews focusing on MSCs and I deleted some lines of this paragraph (page 1, lines 34-37). However, in view of the informative purpose of a review which can have readers not familiar with the topic, an introduction can be of help. I tried to briefly provide consolidated general information flanked by an updated introductory context.

4. The genetic alterations of MSC should be better described: what they consist of?Author should be a little bit less general (i.e. DNA proliferation and/or alteration). It is important not only to provide information but mainly to challenge the reader about a specific hypothesis. What does the author reckon regarding the genetic instability? What can determine this long term effect in MSC? It is only a matter of in vitro manipulation of the cells or the interaction with the microenvironment that it should be taken into account? The genetic instability has been also described for the homologue use of MSC?

A sentence was added (page 3, lines 132-133) with a short list of possible genetic alterations occurring during cell culture. As stated above, genetic alterations can be present in the donor (see the previous response) and expanded in culture  or acquired in culture due to repetitive DNA replications (described in paragraph 3 and Figure 1). The homologous or heterologous use of MSCs can not affect MSC genetic stability, but only the recipient response to these cells in terms of immune response to MSC treatment, cell engraftment and permanence.

5.The sole observation of cytogenetic aberrations in culture of MSC can necessarily imply a tumorigenic effect in vivo? The genetic instability only represents a safety issue or it causes clinical failure? Additional non tumorigenic effects such as alteration of the immune response or lacking of differentiation capacity of MSC can occur? Author has separately discussed senescence and tumorigenesis. Perhaps, these two paragraphs could be merged by clarifying this aspect

In general, as stated in the manuscript, to date no sufficient comprehensive data are available to know all possible alterations. Even more relevant is that functional effects of occurring genetic alterations in in vitro expanded MSCs are largely unknown (discussed in paragraph 5 and 6). Actually, cytogenetic aberrations and other types of DNA damage do not necessarily imply a tumorigenic effect (see paragraphs 3, 5, 6). In vitro data suggest that altered cells can reduce their potency, and it is well known that genetic alterations are an hallmark of cancer, therefore they must be evaluated with caution. Anyway, altered cells are usually not administered. Before clinical use, tests are recommended to check for genetic stability of these cells (discussed in paragraph 7) and damaged cells must be discarded.

6.Besides, standardized methodologies to assess genetic instability of MSC are currently employed in regenerative applications as readout of the clinical outcome? Or they only represent an ancillary quality control? Some are described at page 9.

In each clinical trial (see for example AdipoA project, clinicaltrials.gov) a set of controls including genetic stability is foreseen as release criterion. Tests are not ancillary but mandatory to adhere to European legislation about safety of MSC treatments. Since cells with genetic alterations must be discarded, tests are not used as a readout of the clinical outcome.

Best regards

Simona Neri

Reviewer 2 Report

This is a nice review on the genetic stability of MSCs for regenerative medicine applications. It includes important parts that are relative to the genetic stability. This review can provide valuable knowledge about the current studies on the main issues in regenerative medicine using MS. There are some minor suggestions:

1. Some abbreviation need full name when they show in the text at the first time.

2. It would be better to generate subtitle for part 6 “Genetic stability”.

Author Response

Dear reviewer,

Thank you very much for your positive comments and suggestions. I am very happy for your favorable evaluation. 

Abbreviations were checked throughout the text and full names were added at first citation when absent. A subheading was generated in paragraph 6 named “6.1. Methodological approaches to assess genetic stability”.

Best regards

Simona Neri

Reviewer 3 Report

The review manuscript by Simona Neri discusses biosafety aspects related to the use of mesenchymal stromal/stem cells (MSCs) for clinical applications. The review first provides a very general background about MSCs and their therapeutic potential emphasizing their use for regenerative medicine. The review later describes the different safety issues that can arise from the clinical use of MSCs, disusing the role of cell senescence and transformation particularly during ex-vivo expansion. Critical aspects of tumorigenicity are also discussed, including the potential transformation of MSCs in-vitro, MSC ability to form tumors in-vivo, and MSC potential to promote tumor development. The different techniques to assess genetic alterations with their advantages and disadvantages are clearly presented. Finally, current European regulatory aspects are discussed. Overall, the review is very well organized, carefully written and pleasant to read. The bibliography was carefully chosen and is up-to date.

It is commonly believed that the use of MSCs for cell therapy posses almost no risks in terms of safety. While this may hold true based on some clinical trials data, there is a need to define a common criteria to balance the risks and benefits of using a specific MSCs therapy as well as to standardize the procedures and interpretation of genetic stability studies. Thus, I think the review is relevant and well-timed.

I made some comments below to help improve the quality of this review:

Page 4, line 163. The author discussed about the senescence, cell cycle arrest, and lost of differentiation and function that usually occurs in MSCs cultures after extended passaging. It is mentioned that these changes depend on donor age, tissue source and culture conditions. One good example is represented by human umbilical cord-derived MSCs which can be grown for up to passage 10 without loosing multipotential mesenchymal progenitors [Sarugaser et al., PLoS One. 2009;4:e6498]. Moreover,  human umbilical cord-derived MSCs (UC-MSCs) showed increased telomerase activity and longer telomeres compared with BM-MSCs [Yannarelli et al., Stem Cells 2013;31:215-20]. These results are logical taking into account the neonatal tissue origin of UC-MSCs. In the last years, UC has been proposed as a richer and more homogeneous source of MSCs for cell therapy.

Page 7, line 280. The author discussed about the potential to form teratomas in ESCs and iPSCs vs MSCs. One biological aspect that may explain the lower risk of tumor formation (if any) of MSCs in comparison with embryonic stem cells or induced-pluripotent stem cells is the epigenetic repression for the expression of the pluripotency factor OCT4 in adult stem cells [Yannarelli et al., Stem Cells 2013;31:215-20].

Page 7, from line 298. The author describes the tumor promoting growth action that MSCs can have in a recipient. There were a lot of controversy in the literature as some studies observed tumor progression and others tumor suppression after MSCs therapy. Apparently most of these differences can be attributed to the activation state of the infused MSCs. As it happens with monocytes/macrophages, MSCs can be activated into two different phenotypes, a pro-inflammatory phenotype (MSC1) or an immuno-suppressive phenotype (MSC2) [Waterman RS et al., PLoS One 2010;5:e10088]. More recently, it was demonstrated that MSC1-based therapy attenuates tumor growth whereas MSC2-treatment promotes tumor growth and metastasis [Waterman RS et al., PLoS One 2012;7:e45590]. It is therefore certainly true that the use of MSCs for regenerative therapies is associated with a tumor promoting activity (as unstimulated MSCs or the MSC2 phenotype are used to promote regeneration). I think it is important to discuss this notion here to better clarify the different roles that MSCs can have in tumor progression.

Page 3, line 98, all the cites should be together.

Page 3, line 119, it should say “approximately 1/106cells in adult bone marrow”.

Page 6, line 242, the cite Torre et al. is numbered as 141, but in the references it has the number 179. However, as it is first cited in this place, it should have been cited as number 100.

Author Response

May 8, 2019

Dear reviewer,

Thank you very much for your valuable comments and recommendations. I am very happy for your positive evaluation. I added some comments to the text following your relevant suggestions. Please, find below the detailed changes made in the manuscript.

 Page 4, line 163. The author discussed about the senescence, cell cycle arrest, and lost of differentiation and function that usually occurs in MSCs cultures after extended passaging. It is mentioned that these changes depend on donor age, tissue source and culture conditions. One good example is represented by human umbilical cord-derived MSCs which can be grown for up to passage 10 without loosing multipotential mesenchymal progenitors [Sarugaser et al., PLoS One. 2009;4:e6498]. Moreover,  human umbilical cord-derived MSCs (UC-MSCs) showed increased telomerase activity and longer telomeres compared with BM-MSCs [Yannarelli et al., Stem Cells 2013;31:215-20]. These results are logical taking into account the neonatal tissue origin of UC-MSCs. In the last years, UC has been proposed as a richer and more homogeneous source of MSCs for cell therapy.

This valuable comment and related references were added to the text (page 5, lines 180-183).

Page 7, line 280. The author discussed about the potential to form teratomas in ESCs and iPSCs vs MSCs. One biological aspect that may explain the lower risk of tumor formation (if any) of MSCs in comparison with embryonic stem cells or induced-pluripotent stem cells is the epigenetic repression for the expression of the pluripotency factor OCT4 in adult stem cells [Yannarelli et al., Stem Cells 2013;31:215-20].

This valuable comment and the related reference were added to the text (page 7, lines 294-296).

Page 7, from line 298. The author describes the tumor promoting growth action that MSCs can have in a recipient. There were a lot of controversy in the literature as some studies observed tumor progression and others tumor suppression after MSCs therapy. Apparently most of these differences can be attributed to the activation state of the infused MSCs. As it happens with monocytes/macrophages, MSCs can be activated into two different phenotypes, a pro-inflammatory phenotype (MSC1) or an immuno-suppressive phenotype (MSC2) [Waterman RS et al., PLoS One 2010;5:e10088]. More recently, it was demonstrated that MSC1-based therapy attenuates tumor growth whereas MSC2-treatment promotes tumor growth and metastasis [Waterman RS et al., PLoS One 2012;7:e45590]. It is therefore certainly true that the use of MSCs for regenerative therapies is associated with a tumor promoting activity (as unstimulated MSCs or the MSC2 phenotype are used to promote regeneration). I think it is important to discuss this notion here to better clarify the different roles that MSCs can have in tumor progression.

This valuable comment and related references were added to the text (page 7, lines 317-325).

Page 3, line 98, all the cites should be together.

Cites were grouped as suggested.

Page 3, line 119, it should say “approximately 1/106cells in adult bone marrow”.

Text was corrected.

Page 6, line 242, the cite Torre et al. is numbered as 141, but in the references it has the number 179. However, as it is first cited in this place, it should have been cited as number 100.

The reference number was corrected.

Best regards

Simona Neri